# Changes in Cystoscopic Findings after Intravesical Hyaluronic Acid Instillation Therapy in Patients with Interstitial Cystitis

**DOI:** 10.3390/diagnostics12082009

**Published:** 2022-08-19

**Authors:** Chia-Ju Lin, Chih-Ku Liu, Hsiao-Yun Hsieh, Ming-Jer Chen, Ching-Pei Tsai

**Affiliations:** Department of Obstetrics and Gynecology, Taichung Veterans General Hospital, Taichung 40705, Taiwan

**Keywords:** interstitial cystitis, bladder pain syndrome, hyaluronic acid, cystoscopy, glomerulation, intravesical instillation

## Abstract

(1) Background: Limited data showed changes in glomerulation in the bladder mucosa of patients with interstitial cystitis (IC) after intravesical hyaluronic acid (HA) bladder infusion. We aimed to investigate the above changes. (2) Methods: Medical records of IC patients were reviewed retrospectively, from January 2010 to October 2019. Patients who had received repeated cystoscopy after intravesical HA treatment were enrolled. The associations of multiple parameters, including the ages, symptoms, initial glomerulation stage, HA doses, and the interval period of repeated cystoscopy between the glomerulation change in the repeated cystoscopy were analyzed. (3) Results: Among the 35 patients, 9 cases (25.7%) showed better glomerulation grades in the repeated cystoscope (Group 1), 20 cases (57.1%) showed the same grades (Group 2), and 6 cases showed worse grades (Group 3). No difference was seen in the initial grades or treatment course among the three groups. The interval periods from the initial to the repeated cystoscopy of Group 1 were longer than Group 2 and Group 3 (*p* = 0.031). Group 3 presents an elder age trend than the other two groups. (4) Conclusion: Intravesical HA repaired bladder glomerulation in a small group of patients with IC. Prolonged treatment has potential benefits, while older age is possibly a negative factor. However, no strong correlation was found between the initial glomerulation grades or changes in glomerulation grades with clinical symptoms.

## 1. Introduction

Interstitial cystitis/bladder pain syndrome (IC/BPS) is characterized by a wide range of hypersensitive bladder symptoms (e.g., bladder pain, frequency, urgency, or nocturia) [1,2,3]. Cystoscopy is the standard diagnostic method, and typical findings include glomerulation and Hunner’s lesions [4,5]. IC/BPS pathogenesis, the mechanism of which remains poorly understood [6], is the key barrier to finding a treatment for the disease. The potential causes of pathogenesis include damage to the glycosaminoglycan (GAG) layer of the bladder urothelium, urine toxins that cause activation of bladder sensory nerves, bladder inflammation, and detrusor fibrosis [7,8]. Pathological findings of bladder biopsies from IC/BPS patients revealed impaired homeostasis in the bladder urothelium, which is associated with chronic inflammation [9].

Intravesical instillation of hyaluronic acid (HA) is a known effective treatment to restore the defect of the GAG layer in bladder mucosa and reduce the symptoms of IC/BPS [10,11,12,13,14,15]. In most reported studies, cyclic HA instillation safely reduces pain associated with IC/BPS and the altered urinary frequency to a lesser degree [11]. Currently, few solid evaluations are available on the treatment outcome; most studies used questionnaires/scales to assess the treatment outcome.

Limited data have been reported regarding the glomerulation changes before and after HA instillations. Sahiner et al., demonstrated in the rat model that a single intravesical HA instillation reduces infiltration of inflammatory cells and bladder inflammation severity [16]. Lv et al. reported that the interleukin-6 levels in rat bladders are a possible cause of IC/BPS, and that the level drops after intravesical administration of HA [17]. Rooney et al. demonstrated that intravesical HA significantly enhances the production of sulfated GAG in the urothelium, without altering tight junction expression, as well as exerting a direct effect on anti-inflammation and increasing epithelial permeability [18]. These findings were obtained from an in vitro model. Clinical studies to evaluate the changes in cystoscopic findings after intravesical HA treatment in IC/BPS patients are still limited, or more focused on bladder capacity [19,20].

In this study, we aimed to investigate glomerulation changes after intravesical HA treatment in patients with various outcomes. Specifically, we determined factors affecting such treatment outcomes regarding urothelium repairment.

## 2. Materials and Methods

### 2.1. Patients and Medical Data

In this retrospective study, we reviewed the medical records of female IC/BPS patients undergoing repeated cystoscopy in a tertiary teaching hospital from January 2010 to October 2019. Thirty-five consecutive patients who were newly diagnosed with IC/BPS were enrolled.

Diagnostic criteria were based on the interstitial cystitis guidelines published by the European Society for the Study of Interstitial Cystitis (ESSIC) [2,3,7,20]. The criteria included characteristic symptoms of urinary frequency, nocturia, pelvic discomfort, and cystoscopic findings. Confusable diseases as the cause of the symptoms must be excluded [5]. Patients with a urinary tract infection, urinary retention, stress urinary incontinence, and whoever received other therapies such as sodium pentosan polysulphate, intravesical onabotulinum toxin A, or platelet-rich-plasma injections were also excluded. A rigid cystoscopy under intravenous general anesthesia was performed by trained gynecology specialists. Hydrodistension was performed using 1000 mL of saline (0.9%) at a hydrostatic pressure of 80 cm until the fluid filling spontaneously stopped. The maximum capacity was the total initial fluid amount minus the residual saline volume. To observe petechiae or hematuria, the bladder was first drained through the cystoscope. Once a Hunner’s lesion or suspicious lesion was noticed, a bladder wall biopsy would be performed to exclude malignancy. Glomerulations were graded according to the scheme of Nordling et al. (a reference of ESSIC): Grade 0: normal mucosa; Grade 1: petechiae in at least two quadrants; Grade 2: large submucosal bleeding; Grade 3: diffuse global mucosal bleeding; Grade 4: mucosal disruption, with or without bleeding/edema [21].

After establishing the diagnosis, intravesical instillation of HA was applied. The protocol of HA instillation was referred to in the study by Hung et al. [10] and Yu et al. [20]. A total of 40 mg of sterile high molecular weight HA (Cystistat™) was applied within bladder infusions once a week for the first 4 weeks, and then once a month for the following 5 months. After a total of 9 doses of therapy, a follow-up outpatient meeting was arranged. The physicians had asked about the effect of the HA instillation and the symptoms during the interval period. The responses of the patients were recorded in the visit notes. The protocol of maintenance instillation was determined by the clinical responses of the patients. If the patients had symptom relief after the HA instillation but still complained about residual or refractory LUTS symptoms within a month, the treatment protocol continued with intravesical HA instillation once a month. For patients who were satisfied with previous HA therapy, the treatment protocol could gradually prolong instillation intervals to 2 months or 3 months, or stop the instillation.

The indication of repeated cystoscopy for most patients is an unsatisfied clinical symptom after HA instillation or flare-up of clinical symptoms after prolonged instillation intervals. However, few patients had repeated cystoscopy accompanied by other gynecological surgery due to medical conditions.

All patient data, charts, operative notes, and follow-up records were reviewed. The age at initial diagnosis, clinical symptoms, grades of glomerulation, bladder capacities, HA instillation treatment protocols, and the interval period of repeated scope were all analyzed. The study protocol was approved by the institutional review board of the hospital (Number CE20169A.).

### 2.2. Statistical Analysis

The continuous data were expressed as mean ± SD, and the categorical data were expressed as numbers and percentages. For categorical variables, we used the chi-square test and Fisher’s exact test. The Kruskal–Wallis test and the Mann-Whitney U test were applied for analysis of non-normal distribution variables. Statistical significance was set at *p* < 0.05. All statistical analyses were performed using SAS software version 9.4 (SAS Institute, Inc., Cary, NC, USA).

## 3. Results

A total of 35 patients received HA intravesical instillation therapy, and repeated cystoscopies were analyzed in this study. The demographics of the patients with IC/BPS are shown in Table 1. The mean age of the patients was 46.3 years, and their mean interval of follow-up cystoscope was 26.9 months. The average HA dose was 1 vial per month (range, 0.6 to 2 vials). The initial presence of glomerulation grades had the following distribution: 11 patients with Grade 1, 10 patients with Grade 2, and 14 patients with Grade 3. Their average bladder capacity was 620 mL (Table 2). According to the records of the outpatient follow-up, most initial lower urinary tract symptoms (LUTS) and pain were relieved temporarily after intravesical instillation of HA treatment (Table 1 and Table 3). At an average 1-year follow-up after the HA therapy, only 10~30% of patients complained about bothersome pain and LUTS (Table 3). However, flare-up of clinical symptoms usually occurred after prolonged instillation intervals, which is our indication for repeat cystoscopy.

Among the first and repeated cystoscopies, different glomerulation grades could be found in the same patient. Based on the changes in glomerulation grades, patients were classified into three groups as follows: Group 1: 9 patients (25.7%) with improved glomerulation grades in the repeated cystoscopies; Group 2: 20 patients (57.1%) showed no grade changes; Group 3: 6 patients (17.1%) showed poorer grades (Table 3). Cystoscopic pictures of patients in each group are shown in Figure 1. The distribution of changes in glomerulation grades is shown in Figure 2.

The average time interval between the initial diagnostic cystoscopy to the repeated cystoscopy was 44.8 months for Group 1, 19.3 months for Group 2, and 25 months for Group 3. The interval was significantly longer in Group 1 (*p* = 0.031) (Table 3). No difference was found regarding monthly doses of HA instillation during the follow-up period (*p* = 0.236). No difference was found across the three groups in the distribution of the initial glomerulation grade (*p* = 0.162), the initial bladder volume capacity (*p* = 0.357), or the follow-up capacity of bladder volume (*p* = 0.978). The progress of the mean bladder capacity increased in Group 1, from 556 mL to 637.5 mL, but no difference was seen in the initial capacity and repeated capacity in Group 2 or Group 3.

Group 3 (the group with poorer glomerulation grades) showed an apparent trend of older ages (mean 55 years old) than the other two groups. However, a significant difference was only found between Group 3 and Group 2 (*p* = 0.037) (Table 4).

## 4. Discussion

In this study, we found that the intravesical HA instillation had repaired bladder mucosa damage in some IC/BPS patients (25.7%, 9/35), but no strong correlation was found between the initial glomerulation grades or changes in glomerulation grades with clinical symptoms. Longer periods of repeated instillations may help repair bladder glomerulation. On the contrary, for older patients, the benefits from HA appeared small in terms of repairing bladder glomerulation.

Despite decades of basic and clinical research, the etiology of IC/BPS remains obscure. The current most accepted theory is that the disease is due to injury or dysfunction of the glycosaminoglycan layer (defensive mucosal lining) covering the urothelium [3]. Restoring the GAG layer is the chief aim targeted by IC/BPS treatments. Bladder instillation of HA is thought to provide direct protection to damaged urothelium from IC/BPS with the production of sulfated GAG and reduced bladder epithelial permeability. Several studies have demonstrated that intravesical instillation of HA reduced the bladder pain caused by interstitial cystitis [10,11] and improved patients’ quality of life [6,12]. However, the treatment outcome is not permanent [6]. Few clinical studies have been performed on changes in cystoscopic findings after intravesical HA treatment on IC/BPS patients, with reports mostly on in vitro models of IC/BPS [16,17,18,19]. In our present study, 9 patients’ (25.7%) follow-up cystoscopy outcomes indicated a decrease in their glomerulation grade (Group 1) at a mean interval of 44.8 months after the initial cystoscopic diagnosis; compared with those patients showing no change in glomerulation grades (Group 2) and those with poorer grades (Group 3), their treatment periods were longer (Table 3).

Several studies on the benefits from long-term HA instillation on recurrent LUTS caused by IC/BPS were reported [10,11,22,23,24,25,26], but the only limited correlation was found between the presentation of symptoms and damage of the bladder mucosa. In the study conducted by Kallestrup et al., a long-term positive impact was shown in IC/BPS patients after treatment for three years [10]. Engelhardt et al., reported that 41.7% (20/48) of their patients had symptoms recur during the first year after initial improvements, and treatment responses persisted during repeated instillation therapy throughout the 5-year treatment period [23]. Cervigni et al., with 12 IC/BPS patients, found a steady improvement in symptoms during a 3-year follow-up with instillations containing HA and chondroitin sulfate (CS) [24]. Based on the above studies and our own, long-term instillations likely have benefits on both the relief of refractory IC/BPS symptoms and the recovery of bladder mucosal integrity.

In our study, most of the grades of glomerulations remained unchanged despite HA treatment (57.1%, 20/35), or even worsened (17.1%, 6/35). No strong correlation was found between initial glomerulation grades, changes in the glomerulation grade, and clinical symptoms. Research for a comparison of cystoscopic findings before and after intravesical instillation treatment has rarely been done. In Shear and Mayer’s study, the cystoscopic findings of IC/BPS patients appeared time-variant after intravesical hydrodistension treatment [27]. As a chronic disease, the appearance of the bladder with interstitial cystitis could change over time. Such temporal variation in the findings with hydrodistension adds to the complexity of studying IC/BPS patients. Other possible causes of discrepancies in the results are differences in the underlying multifactorial etiology of IC/BPS [25], and hence intravesical therapy may not always be effective [10,27]. Although the symptoms might not always be related to the cystoscopic findings, the normalized morphology of bladder mucosa indicated a therapeutic tissue repair and regeneration effect. Recently, the potential applications of regenerative medicine, such as platelet-rich plasma (PRP), stromal vascular fraction (SVF), and stem cells, have been shown to be beneficial to treatments for IC/BPS in some preclinical and a few clinical studies [28,29]. The remarkable normalization of bladder mucosal morphology after such therapies has been reported [29]. The discussion could lead to further study ideas in the future.

Regarding possible factors influencing treatment outcomes, we found that our patients in Group 3 had poorer glomerulation grades in response to HA treatment, and they had apparently older ages (mean 55 years old). Advanced age could therefore be a negative factor affecting the treatment outcome on restoring bladder mucosa. Kim et al. reported poor responses to HA compared with the literature. The discrepancy could be related to the refractoriness of their patients to conventional therapy. Their cohort was also older (mean age 57.0 years) than other studies. One possible explanation is that after extended periods of refractory therapy, fibrosis of bladder mucosa had occurred with chronic inflammation, which is also associated with increased urinary frequency and decreased bladder capacity [30]. Despite the fact that fibrosis of the bladder is not a factor evaluated in our study, the role of age and fibrotic changes in repairing bladder mucosa cannot be ruled out; further study will be needed.

No adverse event for HA instillation was found in our patients. That is consistent with most previous reports, except one mentioning a mild bladder irritation [31]. There are several limitations of our study. First, this series only retrospectively analyzed 35 patients with IC/BPS who underwent repeated cystoscopy. The evaluation of the response to HA was compromised by the retrospective nature of the study and the frequent need for additional adjustments in concomitant therapy [27]. The case number is relatively small. Additionally, no standardized questionnaire was completed prior to each cystoscopy. As a result of a long-term retrospective study, the timeline of follow-up outpatient visits and cystoscopies was not standardized. The records of recurrent pain and LUTS were objectively narrated by the patients. Further designed studies on larger databases would strengthen our present findings.

In conclusion, intravesical HA treatment has potential long-term benefits in healing glomerulation of the bladder mucosa, but only in a small portion of patients. There is no strong correlation between changes in glomerulation grade and clinical symptoms. The estimated time required was not clarified yet, and the elder age has a negative impact. Since there is a lack of a model for the bladder mucosa healing process, more basic and clinical research is required.

## 5. Conclusions

HA instillation had repaired bladder mucosa damage in a small portion of IC/BPS patients. No strong correlation was found between the initial glomerulation grades or changes in glomerulation grades with clinical symptoms. Longer periods of repeated instillations may help repair bladder glomerulation. For older patients, the benefits from HA appeared small in terms of repairing bladder glomerulation.

## Figures and Tables

**Figure 1 diagnostics-12-02009-f001:**
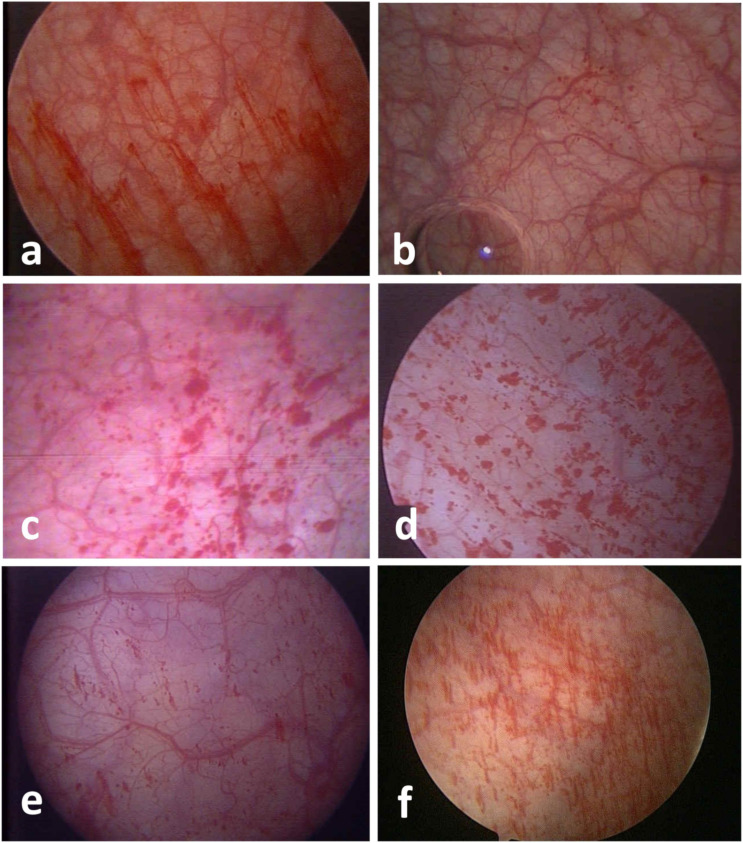
Different changes of glomerulation grades in three patients before and after intravesical HA treatment in cystoscopies. (**a**) A 63-year-old woman with Grade 2 glomerulation of the bladder before intravesical HA treatment. (**b**) The glomerulation grade of patients from (**a**) shifted to Grade 1 after intravesical HA treatment. (**c**) A 53-year-old woman with Grade 3 glomerulation of the bladder before intravesical HA treatment. (**d**) The glomerulation grade remained at Grade 3 in the patient in picture (**c**) after intravesical HA treatment. (**e**) A 42-year-old woman with Grade 2 glomerulation of the bladder before intravesical HA treatment. (**f**) The glomerulation grade worsened to Grade 3 in the patient in picture (**e**) after intravesical HA treatment.

**Figure 2 diagnostics-12-02009-f002:**
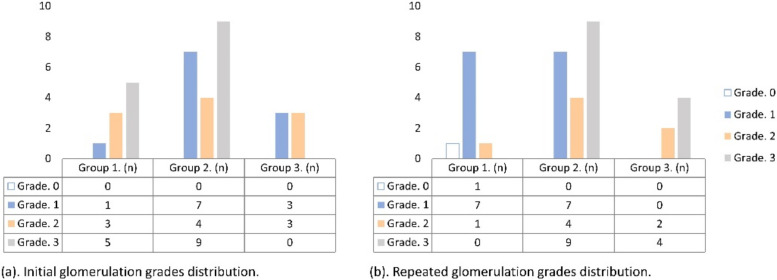
The distribution of glomerulation grades in the initial and follow-up cystoscopic findings in the three groups of patients. (**a**) Distribution of initial glomerulation grades in the three groups. (**b**) Distribution of follow-up glomerulation grades in the three groups.

**Table 1 diagnostics-12-02009-t001:** Basic characteristics of patient (n = 35) who underwent intravesical hyaluronic acid instillation therapy after cystoscopy.

General Data	Value (Mean ± SD)	Range
Mean age (years)	46.3 ± 12.3	18–68
Follow-up interval (months)	26.9 ± 19.9	4.8–72.9
Total HA dosage (vial)	24.5 ± 16.2	4–61
Average HA dosage per month (vial/months)	1.0 ± 0.3	0.6–2

HA, hyaluronic acid.

**Table 2 diagnostics-12-02009-t002:** The data of initial and follow-up cystoscopic findings of the patients (n = 35) who underwent intravesical hyaluronic acid instillation.

	Initial Cystoscopy	Follow-Up Cystoscopy	*p* Value
Max capacity (mL, Mean ± SD)	620.6 ± 143.6	650 ± 130	0.135 *
Glomerulation grade (n)			0.196 #
Grade 0	0 (0%)	1 (2.9%)	
Grade 1	11 (31.4%)	14 (40.0%)	
Grade 2	10 (28.6%)	7 (20.0%)	
Grade 3	14 (40.0%)	13 (37.1%)	
Hunner’s lesion	1 (2.9%)	1 (2.9%)	1.000 **
Terminal hematuria	14 (40.0%)	13 (37.1%)	1.000 **
Trabeculum	23 (65.7%)	18 (51.4%)	0.180 **
Pain	25 (71.4%)	22 (62.9%)	
LUTS	29 (82.9%)	17 (48.6%)	

LUTS, lower urinary tract symptoms. * Willcoxon test, # chi-square test, ** McNemar test.

**Table 3 diagnostics-12-02009-t003:** The data of the three groups with different glomerulation grade changes revealed by cystoscopy after intravesical hyaluronic acid instillation therapy.

	Group 1 (n = 9)		Group 2 (n = 20)		Group 3 (n = 6)		*p* Value
	Improved		Similar		Worsened		
Mean age	45.3	±13.6	44.2	±12.2	55	±7.5	0.121
Mean initial capacity (mL)	555.6	±164.8	642.1	±131.5	650	±141.4	0.357
Mean follow-up capacity (mL)	637.5	±176.8	652.6	±126.4	660.0	±54.8	0.978
Mean follow-up time (months)	44.8	±22.7	19.3	±12.4	25.1	±21.9	0.031 *
Mean HA dose (vial)	35.9	±18.8	19.9	±11.5	23.0	±20.1	0.095
Mean average HA dose(vial/m)	0.9	±0.3	1.1	±0.3	1.0	±0.3	0.236
Before HA pain	9	(100%)	14	(70%)	2	(33.3%)	
LUTS	7	(77.78%)	17	(85%)	5	(83.3%)	
HA for 1 year pain	1	(11.1%)	6	(31.6%)	1	(16.7%)	
LUTS	2	(22.2%)	5	(26.3%)	1	(16.7%)	
Before follow-up cystoscope pain	6	(66.7%)	12	(63.2%)	4	(66.7)	
LUTS	5	(55.6%)	10	(50%)	2	(33.3%)	

Chi-square test. Kruskal-Wallis test, * *p* < 0.05.

**Table 4 diagnostics-12-02009-t004:** The data comparison between the groups with the similar cystoscopic outcomes and worsened outcomes.

	Group 2 (n = 20) Similar		Group 3 (n = 6) Worsened		*p* Value
Mean age	44.2	±12.2	55	±7.5	0.037 *
Mean initial bladder capacity (mL)	642.1	±131.5	650	±141.4	0.831
Mean follow-up time (months)	19.3	±12.4	25.1	±21.9	0.689
Mean HA dose (vial)	19.9	±11.5	23.0	±20.1	0.822
Mean average HA dose (vial/m)	1.1	±0.3	1.0	±0.3	0.487
Mean follow-up bladder capacity(mL)	652.6	±126.4	660.0	±54.8	0.836

HA, hyaluronic acid. Chi-square test. Fisher’s exact test. Mann-Whitney U test. * *p* < 0.05.

## Data Availability

The data presented in this study are available upon request after obtaining additional permission from the Institutional Review Board of Taichung Veterans General Hospital.

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
