# Peer review of "Changes in Cystoscopic Findings after Intravesical Hyaluronic Acid Instillation Therapy in Patients with Interstitial Cystitis"

_diagnostics, 2022, doi:10.3390/diagnostics12082009_

Round 1

Reviewer 1 Report

I think that the article is complete and well written, background, methods and results are well explained and conclusions are supported by reported results.

Article is suitable for publication in the current form

Author Response

We thank the reviewer for the reading of the manuscript and the constructive remarks. We have taken the comments on board to improve and clarify the manuscript.

Reviewer 2 Report

There are some arguements on the role of glomerulation for the diagnosis and evaluation of the theraputic effect of IC/BPS. It is known that there is no relation between the severtity of the disease and the cystoscopic findings. Therefore, the significance of this research is much less importance.

The measurement method of the bladder volumn is not acceptable. The average volumn is more than 600ml, which is also not consistent with clinic findings. The evaluations of the symptoms preoperatively and postoperatively are not enough.

Author Response

Dear Reviewer: 

We appreciate you and the reviewers for your precious time in reviewing our paper and providing valuable comments. It was your valuable and insightful comments that led to possible improvements in the current version. Below we provide the point-by-point responses. 

Response to Reviewer 2 Comments

Point 1:

There are some arguements on the role of glomerulation for the diagnosis and evaluation of the theraputic effect of IC/BPS. It is known that there is no relation between the severtity of the disease and the cystoscopic findings. Therefore, the significance of this research is much less importance.

Response 1:

Although this paper also proved that the symptoms might not be related to mucosa repairment, we demonstrated that intravesical HA do repair bladder glomerulation in some patients. Prolonged HA treatment has potential benefits to glomerulations, while elder age is possibly a negative impact.

Currently, limited data have been reported about the morphological changes before and after HA instillations. The normalization of bladder mucosa potentially has positive impact for the patients, while further studies are necessary to determine whether the clinical observations can be correlated with symptoms before and after treatment.

Point 2

The measurement method of the bladder volumn is not acceptable. The average volumn is more than 600ml, which is also not consistent with clinic findings. The evaluations of the symptoms preoperatively and postoperatively are not enough.

Response 2:

According to the criteria of the ESSIC, hydrodistension must take place under anesthesia, at a pressure of 80 to 100 cmH2O, lasting 1 to 2 minutes[1]. In anethesia condition, the maximal bladder capacity is bigger than awanken condition. To avoid complications, most urologists suggested not to exceed 1,000 mL of infusion.[2]

Hung et al[3]. had demonstrated that the Mean anesthetic bladder capacity was  547.2 ± 207.8ml of patients with insterstitial cystitis responed to intravesical HA treatment. The study of Yu et al. demonstrated that the maximal bladder capacity could be 746±168 ml in patients with grade I glomerulation and 593±197 ml in grade III patients[4].

This series only retrospectively analyzed 35 patients    who underwent repeated cystoscopy.  No standardized questionnaire was done prior to each cystoscopy. The records of pain and LUTS were subjectively narrated by the patients.

  1. van de Merwe JP, Nordling J, Bouchelouche P, et al. Diagnostic criteria, classification, and nomenclature for painful bladder syndrome/interstitial cystitis: an ESSIC proposal. Eur Urol 2008;53:60-7.
  2. Turner KJ, Stewart LH. How do you stretch a bladder? A survey of UK practice, a literature review, and a recommendation of a standard approach. Neurourol Urodyn 2005;24:74-6.
  3. Hung MJ, Tsai CP, Lin YH, Huang WC, Chen GD, Shen PS: Hyaluronic acid improves pain symptoms more than bladder storage symptoms in women with interstitial cystitis. Taiwan J Obstet Gynecol. 2019, 58:417-422. 10.1016/j.tjog.2018.11.033
  4. Yu WR, Jhang JF, Ho HC, Jiang YH, Lee CL, Hsu YH, Kuo HC: Cystoscopic hydrodistention characteristics provide clinical and long-term prognostic features of interstitial cystitis after treatment. Sci Rep. 2021, 11:455. 10.1038/s41598-020-80252-x

Yours sincerely,
Chia-Ju Lin
Corresponding author:
Ching-Pei Tsai, M.D. 

Reviewer 3 Report

The authors analyzed the relationship between glomerulation grades and clinical symptoms pre- and post-treatment to interstitial cystitis using hyaluronic acid (HA) in bladder. There is no significant correlation between them was identified. As author discussed, the multiple limitations in this study weakened the conclusion.

1. It is not clear what the criteria for the follow-up cystoscope is? Is it standard for all enrolled patients? The improved group had the longest treatment duration and highest HA dose (table 3). Would the influence of time and dosage of HA treatment introduce the bias into the analysis? In other word, if the other two groups of patients had the same dose of HA treatment, would they be grouped into the improved group? Furthermore, authors declared that the cystoscopic findings appeared time-variant after treatment. How did the authors to limit this bias since the follow-up cystoscope timing is different to each patient?

2. In table 3, 16 patents had LUTS before the follow-up cystoscope, but in table 2, there are 17 patients with LUTS. 

3. The statistical method for table 2 and 3 are not clear. In table 2, only the p value for the continuous variable using Kruskal Wallis test. However, there is no p value for the Chi-Square test.  In table 3, the * symbol corresponds to two different types of statistical tests and there is no data suitable for Chi-Square test. 

Minor points

1. there are some typos in the context. For example, what do the numbers (107 in line 104 and 108 in line 142) represent? 

Author Response

Dear Reviewer:

We appreciate you and the reviewers for your precious time in reviewing our paper and providing valuable comments. It was your valuable and insightful comments that led to possible improvements in the current version. The authors have carefully considered the comments and tried our best to address every one of them. Below we provide the point-by-point responses. 

Point 1: It is not clear what the criteria for the follow-up cystoscope is? Is it standard for all enrolled patients? The improved group had the longest treatment duration and highest HA dose (table 3). Would the influence of time and dosage of HA treatment introduce the bias into the analysis? In other word, if the other two groups of patients had the same dose of HA treatment, would they be grouped into the improved group? Furthermore, authors declared that the cystoscopic findings appeared time-variant after treatment. How did the authors to limit this bias since the follow-up cystoscope timing is different to each patient?

 Response 1:

For a retrospective study, to organize a standardized protocol would be relatively difficult. Though the following time of cystoscopy is not regular, we aimed to investigate if there were similar features between the patients with better bladder mucosa recovery or worse conditions. The improved group actually had the longest treatment duration and highest HA dose. Therefore, we concluded that prolong instillitation could be a positive feature and older age could be a negative feature. However, the evidence is not sufficient yet. Further prospective studies will be needed.

Although the symptoms might not always be related to the cystoscopic findings, the normalized morphology of bladder mucosa indicated a tissue repair and potential regenerative effect. This is why we want to report this paper.

Point 2: In table 3, 16 patents had LUTS before the follow-up cystoscope, but in table 2, there are 17 patients with LUTS. 

Response 2:

We are deeply appreciative of the careful check. There were some mistakes that happened when copying the previous data. We will make corrections and check repeatedly in the new version.

Point 3: The statistical method for table 2 and 3 are not clear. In table 2, only the p value for the continuous variable using Kruskal Wallis test. However, there is no p value for the Chi-Square test.  In table 3, the * symbol corresponds to two different types of statistical tests and there is no data suitable for Chi-Square test. 

Response 3:

There are three groups listed for comparison. We used the Kruskal Wallis test for continuous variables such as age, capacity, and time interval. The Chi-Square test was used for the non-continuous variables, such as the patient numbers in pain and LUTS Categories.

Point 4: Minor points

 there are some typos in the context. For example, what do the numbers (107 in line 104 and 108 in line 142) represent? 

Response 4:

We have truly appreciated the clear checking. We had updated the correction in the new version.

 Yours sincerely,
Chia-Ju Lin M.D.
Corresponding author:
Ching-Pei Tsai, M.D. 

Round 2

Reviewer 2 Report

Because there is no relationship between the severtity of IC/BPS and the cystoscopic findings, the significance of this research is much less importance. 

The clinical datas of the patients are not sufficient for the assessment of disease before and after the treatment. 'Hunner's ulcer" should be replaced by "hunner's lesion’. In the manuscript line 153-154, the expression ‘ 25 months 108 for Group 3' is not correct.

Author Response

Dear Reviewer,

Thank you for your precious time in reviewing our paper and providing valuable comments.. Our responses are below:  

Point 1:

Because there is no relationship between the severtity of IC/BPS and the cystoscopic findings, the significance of this research is much less importance.? The clinical datas of the patients are not sufficient for the assessment of disease before and after the treatment.

Response 1:

Previously, limited data showed the changes in glomerulation in bladder mucosa of patients with interstitial cystitis (IC) after intravesical hyaluronic acid (HA) bladder infusion, therefore we aimed to investigate the above changes. For a retrospective study, to organize a standardized protocol would be relatively difficult. Though the following time of cystoscopy is not regular, we aimed to investigate if there were similar features between the patients with better bladder mucosa recovery or worse conditions. Prolong instillitation could be a positive feature and older age could be a negative feature, but the evidence is not sufficient yet.

Although the symptoms might not always be related to the cystoscopic findings, the normalized mor-phology of bladder mucosa indicated a tissue repair and regeneration therapeutic effect. Recently, the potential applications of regenerative medicine, such as platelet-rich plasma (PRP), stromal vascular fraction (SVF) and stem cells, have been shown to be beneficial to treatment of IC/BPS in some preclinical and few clinical studies [1,2]. The remarkable normalization of bladder mucosal morphology after such therapies had been reported [2]. The discussion could lead to further study ideas in the future. We had added this portion in the new manuscript.

Point 2

'Hunner's ulcer" should be replaced by "hunner's lesion’. In the manuscript line 153-154, the expression ‘ 25 months 108 for Group 3' is not correct.

Response 2:

We are truly appreciated for the clear checking and will make corrections.

1.Lin CC, Huang YC, Lee WC, Chuang YC. New frontiers or the treatment of interstitial cystitis/bladder pain syndrome - focused on stem cells, platelet-rich plasma, and low-energy shock wave. Int Neurourol J 2020;24:211–21.

2.Hung MJ, Tsai CP, Ying TH, Chen GD, Su HL, Tseng CJ: Improved symptoms and signs of refractory interstitial cystitis in women after intravesical Nanofat plus platelet-rich plasma grafting: A pilot study. J Chin Med Assoc. 2022, 85:730-735. 10.1097/jcma.0000000000000735

Sincerely, 

Chia Ju Lin

Reviewer 3 Report

Accept in present form. 

Author Response

Dear Reviewer,

We appreciate for your precious time in reviewing our paper and providing valuable comments. Your valuable and insightful comments had led to possible improvements in the current version.

ChiaJu Lin 

This manuscript is a resubmission of an earlier submission. The following is a list of the peer review reports and author responses from that submission.

Round 1

Reviewer 1 Report

I had read the manuscript entitled: "Changes in cystoscopic findings after intravesical hyaluronic acid instillation therapy in patients with interstitial cystitis." This study shows the results of a retrospective study of 35 patients with interstitial cystitis. The authors analyze the changes in cystoscopic findings after intravesical hyaluronic acid instillation. 

The results can help some patients to improve their health. However, this paper can improve. 

Materials and Methods explain the treatment procedure, but the authors also must explain how they formed the initial groups because it is unclear how they grouped them. Line 101 says, "Based on the changes in glomerulation grades from the follow-up cystoscopy, patients were separated into three groups as follows..." Nevertheless, these changes were observed in repeated glomerulation grades distribution

I suggest that they specify if they use standard error or standard deviation in statistical analyses. 

Describe column 2 of Table 2 to understand how the groups were obtained after follow-up cystoscopy.

Table 4 compares group 2 with group 3. The data in this table are also expressed in table 3. I wonder why the standard error or standard deviation of all the data in group 2 is different from that shown in table 3. With the standard error shown in Table 3 for the age of group 2,  is there a significant difference compared with group 3?

The authors should make an effort to explain the pain and LUTS results in the discussion. There does appear to be a significant change in these two symptoms after one year of HA treatment. 

Author Response

Response to Reviewers

Dear Editors:

Thank you for the opportunity to revise our manuscript, Changes in cystoscopic findings after intravesical hyaluronic acid instillation therapy in patients with interstitial cystitis. We appreciate the careful review and constructive suggestions. We believe that the manuscript is substantially improved after making the suggested edits. Below we provide the point-by-point responses. Changes made in the manuscript are marked using track changes. We hope the manuscript after careful revisions meet your high standards. The authors welcome further constructive comments.

Sincerely,

Chia-Ju Lin, MD.

Response to Reviewer 1 Comments

Point 1: Materials and Methods explain the treatment procedure, but the authors also must explain how they formed the initial groups because it is unclear how they grouped them. Line 101 says, "Based on the changes in glomerulation grades from the follow-up cystoscopy, patients were separated into three groups as follows..." Nevertheless, these changes were observed in repeated glomerulation grades distribution

Response 1: We added some clarification of how the group formed in the result, starting from Line 101. As a retrospective study, the idea of the group classification was based on observations. The groups are formed according to the changes in glomerulation grades in the initial and repeated cystoscopies. Patients with lower grades in the repeated cystoscopy than in the initial cystoscopies, which means their bladder mucosa had recovered from glomerulation after HA instillation, were classified into group 1. The patients with no change in the start and repeat cystoscopies were in group 2, and the patients with worse bladder mucosa condition after HA treatment were in group 3. There are no correlations between initial grades and the outcomes.

Point 2: I suggest that they specify if they use standard error or standard deviation in statistical analyses. 

Response 2: We agree with the reviewer and have made adjustments in Table 1 to Table 4.

Point 3: Describe column 2 of Table 2 to understand how the groups were obtained after follow-up cystoscopy. Table 4 compares group 2 with group 3. The data in this table are also expressed in table 3. I wonder why the standard error or standard deviation of all the data in group 2 is different from that shown in table 3. With the standard error shown in Table 3 for the age of group 2,  is there a significant difference compared with group 3?

Response 3: We felt sorry that column 2 in Table 4 (group 2) was mistakenly put in the data from column 4 of Table 4. We are deeply appreciated to the reviewer’s detailed help and have made corrections to the data.

Point 4: The authors should make an effort to explain the pain and LUTS results in the discussion. There does appear to be a significant change in these two symptoms after one year of HA treatment. 

Response 4: We added a description of the previous findings of the effect of pain and LUTS from previous studies in Line 131.

Reviewer 2 Report

Interstitial cystitis represent a chronic condition causing LUTS and bladder pain.

Major issues:

  • no objective measurement of pain (ex: VAS scale) or standardized questionnaire for interstitial cystitis  (ex: O'Leary Sant questionnaire).
  • hydrodistention is usually performed under regional or general anesthesia in order to decrease bladder muscle contractions. The authors should state why the cystoscopy was performed under intravenous anesthesia
  • Cystoscopy was perfomed with flexible or rigid cystoscope?
  • page 2 line 75: " inadequate response or flare-up of clinical symptoms after prolong instillation interval". Please define inadequate response, flare up and time frame of "prolonged interval"
  • page 2 line 79 and page 2 line 57: Patients with urinary tract infection, urinary retention, and stress urinary incontinence were excluded" and  "The patients who had used other therapy such as sodium pentosan polysulphate, intravesical onabotulinumtoxin A, or platelet rich-plasma injections during the study period were excluded" - please create a single section for exclusion criteria 
  • Please specify if a biopsy was perfomed during cystoscopy for confirmation or IC/PBS or to exclude CIS
  •  

Author Response

Response to Reviewers

Dear Editor:

Thank you for the opportunity to revise our manuscript, Changes in cystoscopic findings after intravesical hyaluronic acid instillation therapy in patients with interstitial cystitis. We appreciate the careful review and constructive suggestions. We believe that the manuscript is substantially improved after making the suggested edits. Below we provide the point-by-point responses. Changes made in the manuscript are marked using track changes. We hope the manuscript after careful revisions meet your high standards. The authors welcome further constructive comments.

Sincerely,

Chia-Ju Lin, MD.

Response to Reviewer 2 Comments

Point 1: No objective measurement of pain (ex: VAS scale) or standardized questionnaire for interstitial cystitis  (ex: O'Leary Sant questionnaire).

Response 1: We had tried to quantize the symptoms of the patient in the study. However, only subjective descriptions were obtained. Due to this being a retrospective study that crossed for 10 years, a standard questionnaire to evaluate LUTS was not able to obtain. The VAS of the patients were not routinely recorded in the OPD visits. We list this point in our limitations either.

Point 2: hydrodistention is usually performed under regional or general anesthesia in order to decrease bladder muscle contractions. The authors should state why the cystoscopy was performed under intravenous anesthesia. Cystoscopy was perfomed with flexible or rigid cystoscope?

Response 2: We added the details in the Materials and methods. The type of cystoscope is a rigid cystoscope (Line 60), and the anesthesia method is intravenous general anesthesia performed by an anesthesia specialist (Line 60).

Point 3: Please specify if a biopsy was performed during cystoscopy for confirmation or IC/PBS or to exclude CIS

Response 3: We added the description of bladder biopsy routine to rule out malignancy. (Line 63)

Point 4: page 2 line 75: " inadequate response or flare-up of clinical symptoms after prolong instillation interval". Please define inadequate response, flare up and time frame of "prolonged interval"

Response 4: We modified the description of “inadequate response “ to “unsatisfied clinical symptoms after HA instillation” and , defined “flare-up of clinical symptoms” as temporary relief (Line 75). The“Prolong instillation interval” indicated the instillation interval which more than 1 month (Line 70).

Point 5: page 2 line 79 and page 2 line 57: Patients with urinary tract infection, urinary retention, and stress urinary incontinence were excluded" and  "The patients who had used other therapy such as sodium pentosan polysulphate, intravesical onabotulinumtoxin A, or platelet rich-plasma injections during the study period were excluded" - please create a single section for exclusion criteria

Response 5: We rearranged the order of descriptions (Page 2 line 79 to line 58).

Round 2

Reviewer 2 Report

The connection between glomerulations and bladder pain syndrome/interstitial cystitis is much debated in the literature since several papers showed glomerulations in asymptomatic populations and that there is no association between dissaperance/worsening of this cystoscopic finding with clinical relief.

The authors failed to improve their methodology. The usage of specific questionnaires is mandatory in evaluating subjective findings such as bladder pain or LUTS. This is not a limitation of the study, it represents the entire “the material and method” section

Page 3/line 55: The criteria included characteristic symptoms of urinary frequency, nocturia, pelvic discomfort – if no questionnaire was used, how did the authors included the patients in their study cohort?

 In page 5 line 99, the authors stated: At average 1 year follow-up after HA therapy, only 10~30% of patients complained about bothersome pain and LUTS (Table3). If no questionnaire was used, how did the authors defined bothersome or satisfaction of the patients or evaluated the worsening of symptoms?

No specific follow-up, no strict criteria for HA maintenance only “complaing” (page 3 line 50: If the patients had symptoms relief after HA instillation but still complained about residual or refractory LUTS”)

Author Response

Response to Reviewer 2 Comments

Point 1 The connection between glomerulations and bladder pain syndrome/interstitial cystitis is much debated in the literature since several papers showed glomerulations in asymptomatic populations and that there is no association between dissaperance/worsening of this cystoscopic finding with clinical relief.

Response 1: The purpose of this retropective study is to report the “cystoscopic findings” after HA therapy. The limited association between disappearance/worsening of cystoscopic findings and clinical symptoms is one of our main conclusion. We will add this into the conclusion period of Abstract. We noted most patients had temporary relief of the symptoms after HA instillation (At average 1 year follow-up after HA therapy, only 10~30% of patients complained about bothersome pain and LUTS). However,their symptoms still may falre up as time went by, which is the indication for repeated cystoscopy. The findings in repeated cystoscopies could be different. Some of the findings showed healing of glomerulation, and some showed worsening of glomerulation. 

Point 2. The authors failed to improve their methodology. The usage of specific questionnaires is mandatory in evaluating subjective findings such as bladder pain or LUTS. This is not a limitation of the study, it represents the entire “the material and method” section

Response 2: Questionaires and bladder diary were routinely used, but not in a regular period. However, the changes in symptoms were recorded in every visit notes. Due to the retrospective design of this study, some questionaire results were missing. It’s hard to demonstrate these data in the study.

Point 3. Page 3/line 55: The criteria included characteristic symptoms of urinary frequency, nocturia, pelvic discomfort – if no questionnaire was used, how did the authors included the patients in their study cohort?

Response 3. According to the AUA guideline amendment in 2015, the diagnosis of interstitial cystitis is established as “An unpleasant sensation (pain, pressure, discomfort) perceived to be related to the urinary bladder, associated with lower urinary tract symptoms of more than six weeks duration, in the absence of infection or other identifiable causes”. Symptoms may either be volunteered or described during the patient interview, which was our diagnosis basis for IC.

Point 4. In page 5 line 99, the authors stated: At average 1 year follow-up after HA therapy, only 10~30% of patients complained about bothersome pain and LUTS (Table3). If no questionnaire was used, how did the authors defined bothersome or satisfaction of the patients or evaluated the worsening of symptoms?

Response 4. The paitents had scheduled reservation of clinic visit. The physicians had asked about the effect of HA instillation and the symptoms during the interval period. The responses of the patients were recorded in the visit note. The above description will be added in the material and method.

Point 5. No specific follow-up, no strict criteria for HA maintenance only “complaing” (page 3 line 50: If the patients had symptoms relief after HA instillation but still complained about residual or refractory LUTS”)

Response 5:

During every visit, the patient would be interviewed about the symptoms during this period. After the first protocol of standard treatment, the interval of HA instillation plan would be adjusted based on the clinical response of individuals. If the patient is satisfied with the treatment effect during the interval period, we discussed prolonged intervals of HA instillation. If the symptoms were temporarily relieved after HA instillation and relapsed before the next instillation, the shortened interval could be considered. Since this is a retrospective study, the patients’ follow-up was not standardized.  
